# Causal Effects of Modifiable Behaviors on Prostate Cancer in Europeans and East Asians: A Comprehensive Mendelian Randomization Study

**DOI:** 10.3390/biology12050673

**Published:** 2023-04-29

**Authors:** Yongle Zhan, Xiaohao Ruan, Pei Wang, Da Huang, Jingyi Huang, Jinlun Huang, Tsun Tsun Stacia Chun, Brian Sze-Ho Ho, Ada Tsui-Lin Ng, James Hok-Leung Tsu, Rong Na

**Affiliations:** 1Division of Urology, Department of Surgery, LKS Faculty of Medicine, The University of Hong Kong, Hong Kong, China; 2Department of Urology, Ruijin Hospital, Shanghai Jiao Tong University School of Medicine, Shanghai 200025, China; 3Department of Statistics, Miami University, Oxford, OH 45056, USA; 4Division of Urology, Department of Surgery, Queen Mary Hospital, Hong Kong, China

**Keywords:** prostate cancer, modifiable lifestyle behavior, causality, mendelian randomization

## Abstract

**Simple Summary:**

As lifestyle intervention has come to the fore for cancer prevention, the causal effect of lifestyle behaviors on prostate cancer remains to be elucidated. No comparison across different ethnicities has been conducted previously. In this study, we used a novel instrumental variable methodology, namely, Mendelian randomization analysis, to explore the associations between a wide range of behavioral factors and prostate cancer in both Europeans and East Asians. The results of this study showed that lifetime tobacco smoking was positively related to an increased prostate cancer risk in Europeans. In East Asians, alcohol consumption and delayed sexual initiation were risk factors, while vegetable intake was a protective factor for prostate cancer. These findings largely broaden the evidence base for the spectrum of risk factors for prostate cancer, and provide insights into the priority of behavioral interventions for high-risk groups in different population settings.

**Abstract:**

Objective: Early evidence is disputable for the effects of modifiable lifestyle behaviors on prostate cancer (PCa) risk. No research has yet appraised such causality in different ancestries using a Mendelian randomization (MR) approach. Methods: A two-sample univariable and multivariable MR analysis was performed. Genetic instruments associated with lifestyle behaviors were selected based on genome-wide association studies. Summary-level data for PCa were obtained from PRACTICAL and GAME-ON/ELLIPSE consortia for Europeans (79,148 PCa cases and 61,106 controls), and ChinaPCa consortium for East Asians (3343 cases and 3315 controls). Replication was performed using FinnGen (6311 cases and 88,902 controls) and BioBank Japan data (5408 cases and 103,939 controls). Results: Tobacco smoking was identified as increasing PCa risks in Europeans (odds ratio [OR]: 1.95, 95% confidence interval [CI]: 1.09–3.50, *p* = 0.027 per standard deviation increase in the lifetime smoking index). For East Asians, alcohol drinking (OR: 1.05, 95%CI: 1.01–1.09, *p* = 0.011) and delayed sexual initiation (OR: 1.04, 95%CI: 1.00–1.08, *p* = 0.029) were identified as risk factors, while cooked vegetable consumption (OR: 0.92, 95%CI: 0.88–0.96, *p* = 0.001) was a protective factor for PCa. Conclusions: Our findings broaden the evidence base for the spectrum of PCa risk factors in different ethnicities, and provide insights into behavioral interventions for prostate cancer.

## 1. Introduction

In 2020, prostate cancer (PCa) became the second most common male cancer and the fifth leading cause of male cancer death, with up to 1.41 million new cases and 0.37 million deaths worldwide [1]. PCa has a significant economic burden due to the cost of treatment, loss of productivity, and increase in caregiver load [2]. Given the unclear consensus on treatment strategies, disputable benefits of prostate-specific antigen (PSA) screening, and invasiveness of predictive biomarkers at present [3], an affordable and practicable preventive measure with no side effects is warranted, among which a healthy lifestyle could be considered a priority [4]. 

Several lifestyle factors have been reported to play an essential role in PCa development. A systematic review with 4 million participants found increased PCa mortality in current cigarette smokers [5]. A dose–response meta-analysis suggested that the consumption of liquor at 14 g/d was associated with a 12% higher aggressive PCa risk [6]. Another meta-analysis, focusing on low- and middle-income countries, observed an inverse association between vegetable intake and risk of PCa [7]. Nevertheless, the results of lifestyle behaviors in relation to PCa were highly inconsistent and controversial between studies and ethnicities. Moreover, it is a challenge to estimate the causal impact of these behaviors from observational studies due to the unmeasured confounders and reverse causation bias. Mendelian randomization (MR) analysis, a methodology based on instrumental variable (IV) principles, is invoked to explore the causality between exposures and outcomes. This approach provides us with the opportunity to perform causal inference between various behavioral factors and PCa risks. For example, several MR studies have examined the causal associations of physical activity (PA) [8], cannabis use [9], tobacco smoking [10], and alcohol drinking with risks of PCa [10]. 

However, most of these MR studies were limited to either a single behavior or a single ancestry. The genetic correlation between tobacco and alcohol use was recently reported by a genome-wide association study (GWAS) [11], suggesting genetic variants to be potentially overlapped among common lifestyle behaviors. In addition, the widely reported PCa genetic differences amid inter-ethnic populations suggested a discrepant spectrum of PCa risk factors [12]. 

Therefore, we performed a comprehensive two-sample univariable and multivariable Mendelian randomization (UVMR, MVMR) analysis regarding the causality between behaviors and PCa in both European and East Asian ancestries, under the assumptions of (i) different lifestyle behaviors being etiologically and genetically related, and (ii) the behaviors–PCa relationship being racially diverse. The aims of this study were to broaden the evidence base for the spectrum of PCa risk factors and to provide insights into behavioral interventions for PCa in different populations. 

## 2. Materials and Methods

The conceptual framework and the three MR assumptions of the current study are illustrated in Figure 1; briefly: (i) instrumental variables (single nucleotide polymorphisms [SNP]) are truly associated with the behavioral factors, (ii) SNPs are unrelated to the confounders on the behaviors–PCa nexus, (iii) SNPs affect PCa only via behavioral factors. SNPs were obtained from summary data from reported GWAS and a GWAS cohort of PCa in the East Asian population (Chinese) [13,14]. The study was reported in accordance with the STROBE-MR guidelines [15]. Informed consent and institutional review board approval were obtained for PCa GWAS in Chinese populations at each hospital, as described in previous studies [13,14].

### 2.1. Selection of Genetic Variants

Seventeen traits of seven types of behavioral factors were selected: tobacco smoking (smoking status, initiation, lifetime smoking index [LSI]), alcohol drinking (frequency, quantity), fruit intake (fresh, dried), vegetable intake (raw, cooked), physical activity (vigorous, moderate, light), sexual behavior (age of initiation, number of sexual partners), and sleep behavior (insomnia, nap during day, daytime dozing). The instrumental variables (SNPs) of these traits were obtained from publicly downloadable sources, including (i) the UK Biobank study, (ii) the GWAS and Sequencing Consortium of Alcohol and Nicotine use (GSCAN), (iii) the MRC Integrative Epidemiology Unit (MRC-IEU), and (iv) the Within family GWAS consortium.

The selection criteria of the SNPs of these traits were: (i) the largest GWAS studies to date, (ii) summative data available and sufficient for our analyses; (iii) clear definition, as well as a relatively accurate evaluation of the magnitude of the exposures. To meet the relevance assumption, we further restricted those SNPs associated with behaviors at a genome-wide significant level (*p* < 5 × 10^−8^). Due to a small number of SNPs of these traits performed in the East Asian population, the *p* level was set at 5 × 10^−5^. In addition, only those with a long physical distance (not within the same linkage disequilibrium region; ≥10,000 kb for European ancestry; ≥5000 kb for East Asian ancestry) and less probable linkage disequilibrium (r^2^ < 0.001 for European ancestry; r^2^ < 0.01 for East Asian ancestry) were retained. To meet the independence assumption, socioeconomic factors and body mass index were empirically assumed to be confounders, and were further excluded by searching the Phenoscanner V2 database (Appendix A). The final included SNPs were listed in Appendix A.

### 2.2. Outcome Traits

GWAS results for PCa in European ancestry were obtained from the Prostate Cancer Association Group to Investigate Cancer-Associated Alterations in the Genome (PRACTICAL) and Genetic Associations and Mechanisms in Oncology/Elucidating Loci Involved in Prostate Cancer Susceptibility (GAME-ON/ELLIPSE) consortia (79,148 PCa cases and 61,106 controls) [16], whereas those for East Asian ancestry were from the Chinese Consortium for Prostate Cancer Genetics (ChinaPCa) (3343 and 3315) [13,14]. Data from FinnGen (6311 and 88,902) and BioBank Japan (BBJ) (5408 and 103,939) were obtained for replication. The characteristics of the GWAS summary data for exposures and outcomes are presented in Appendix A.

### 2.3. Statistical Analysis

The random-effects inverse-variance weighted (IVW) method was used to combine the effects of proxy SNPs on prostate cancer [17]. Cochran’s Q test with I^2^ statistics (low: <25%, moderate: 25~75%, high: >75%) was used to detect heterogeneity amid SNPs. Instrument strength was evaluated with the F-statistic, with a value of >10 being considered sufficient. The R^2^ was calculated to explain the total proportion of phenotypic variance by all included SNPs. The statistical power was calculated via the mRnd platform (https://shiny.cnsgenomics.com/mRnd/, accessed on 1 October 2022). The magnitude of bias due to participant overlap was estimated through a web application developed by Burgess et al. (https://sb452.shinyapps.io/overlap/, accessed on 1 October 2022). 

The horizontal pleiotropy (the variant affecting disease outside of its effect on the exposure) was assessed by the MR-Egger intercept. Outlier detection and sensitivity analyses were performed to check whether the results were robust. Finally, a multivariable MR analysis was applied to estimate the effect of multiple exposures with shared genetic predictors on the risk of PCa [18]. Taking the collinearity of multiple exposures into account, we only selected one phenotype with strongest effect from each behavior in the MVMR models. An additional sensitivity analysis of MVMR was performed using the MR-least absolute shrinkage and selection operator (LASSO) method. A detailed analytic plan for this study was described in Appendix A.

All statistical tests were two-tailed, and associations were considered statistically significant at a *p* < 0.05 level. The analyses were performed using the TwoSampleMR (v0.5.6), MendelianRandomization (v0.6.0), MRPRESSO (v1.0), RadialMR (v1.0), and MVMR (v0.3) packages in R4.1.2 (R Foundation for Statistical Computing, Vienna, Austria).

## 3. Results

### 3.1. Baseline Characteristics

Detailed information on the exposure-associated SNPs for MR analyses in European ancestry was shown in Table 1. Briefly, the number of SNPs ranged from 3 (PA) to 169 (sexual initiation), and the proportions of variance explained by SNPs (R^2^) varied from 0.01% (PA) to 2.44% (smoking initiation). The F statistics for most traits exceeded 10, suggesting no potential weak instrument bias. The power to detect a significant effect size (OR = 0.8/1.2) in our MR analyses was highest for smoking initiation (>99% for discovery and 68% for replication) and alcohol drinking frequency traits (>99% for discovery and 70% for replication), while lowest for PA traits (5% for both discovery and replication). Due to the low rate of sample overlap (<10%) between exposure and outcome datasets, the maximum bias caused by sample overlap was deemed negligible (bias estimate < 0.005). 

For East Asian ancestry (Table 2), the number of SNPs ranged from 3 (PA) to 19 (cooked vegetable intake), and the R^2^ varied from 0.36% (raw vegetable intake) to 7.56% (cooked vegetable intake). A considerable weak instrument bias was detected in most traits (F-stat < 10). The power to detect a significant effect size was highest for the cooked vegetable intake trait (71% for discovery and 98% for replication), while lowest for the raw vegetable intake trait (9% for discovery and 14% for replication). Potential bias caused by sample overlap was absent due to the absence of sample overlap between the exposure and the outcome dataset.

### 3.2. MR Analyses in Europeans

In the UVMR analysis, lifetime smoking (OR: 0.75, 95%CI: 0.58–0.97, *p* < 0.001) and alcohol drinking frequency (OR: 0.89, 95%CI: 0.80–0.99, *p* = 0.038) were significantly associated with lower PCa risks, while an older age at first sexual activity was associated with higher risks (OR: 1.18, 95%CI: 1.01–1.38, *p* = 0.035) (Appendix A, Figure 2). The heterogeneity among SNPs of most traits was low-to-moderate (Appendix A), but the results remained similar in the outlying exclusion analyses (Appendix A). The MR-Egger intercepts indicated limited levels of horizontal pleiotropy in all IVs (Appendix A). The effects of each single SNP were shown in Appendix A.

Conversely, in the MVMR analysis, we found a potential causal effect of lifetime smoking on the risk of PCa (OR: 1.95, 95%CI: 1.09–3.50, *p* = 0.027 per addition standard deviation (SD) increase in the LSI) after adjusting for the seven behaviors (Figure 2). The result was robust in the MR-LASSO analysis (Appendix A). We further replicated the UVMR and MVMR analyses for Europeans in the FinnGen summary dataset, but did not observe a consistent result (Appendix A).

### 3.3. MR Analyses in East Asians

In the UVMR analysis, increased alcohol frequency was significantly associated with higher PCa risk (OR: 1.11, 95%CI: 1.00–1.23, *p* = 0.050) (Appendix A, Figure 3). The Q statistics and MR-Egger intercepts indicated low heterogeneity and limited evidence of directional pleiotropy amid all SNPs of most traits (Appendix A). The plots to inspect the effects of each single SNP on PCa were presented in Appendix A.

In the MVMR analysis, we found a possible causal relationship of alcohol drinking frequency (OR: 1.05, 95%CI: 1.01–1.09, *p* = 0.011) and delayed sexual initiation (OR: 1.04, 95%CI: 1.00–1.08, *p* = 0.029) with higher PCa risks. We also found a potential causal relationship between cooked vegetable intake and lower PCa risks (OR: 0.92, 95%CI: 0.88–0.96, *p* = 0.001) (Figure 3). The result was robust in the MR-LASSO analysis (Appendix A). We further replicated the UVMR and MVMR analyses for East Asians in the BBJ summary dataset, but did not observe a consistent result (Appendix A).

## 4. Discussion

To the best of our knowledge, this was the first MR study to apply both univariable and multivariable MR approaches to systematically determine the causal effects of a series of modifiable behaviors on prostate cancer in two different ancestries. In general, we found significant evidence that lifetime smoking was a risk factor for PCa in Europeans; alcohol drinking and delayed sexual initiation were risk factors, whereas cooked vegetable consumption was a protective factor for PCa in East Asians. These findings provided insights into the prevention and treatment of PCa by the early identification of and intervention in modifiable lifestyle factors. 

Tobacco smoking is an established risk factor and is reported to cause at least 18 types of human cancers with sufficient evidence [19]. However, PCa was not on the list and the latest pooled studies even found an inverse association on the smoking–PCa nexus [20]. Such a finding is sometimes attributed to detection bias [21]. However, in a recent univariable MR study, such a concordant association was still observed [10]. The prior explanation might be inapplicable, and we thus proposed two potential explanations for such an irrational finding. Firstly, smoking behavior should be a synthesis of smoking status and smoking duration, heaviness, and cessation. Using a single dimension as the exposure could bias the true effect. Secondly, smoking strongly co-existed with other lifestyle factors, and these behaviors were assumed to be etiologically and genetically related. A biased estimate might be generated when applying the univariable MR approach. Given the above presumptions, we used the lifetime smoking index to capture a comprehensive smoking behavior picture [22], and applied the multivariable MR approach to infer the causal effect of tobacco smoking on PCa. Our MVMR result indicated that each additional SD increment in the lifetime smoking index could increase a 95% risk of incident PCa. From a biological perspective, smoke-induced carcinogens, DNA methylation, elevated testosterone level, prostatic inflammation, etc., are involved in the PCa development and progression [4]. Based on the established mechanisms and the supportive evidence from our study, we purported a firm causality between tobacco smoking and PCa risk.

The dispute regarding alcohol consumption and PCa risk is also longstanding. Previous research conducted on Europeans did not support the evidence of alcohol consumption increasing PCa incidence [10]. On the other hand, in East Asians, increasing evidence revealed a potential etiological role of alcohol on PCa [23]. This is the first MR study to verify this causality in East Asians. An underlying explanation for this disparity among populations is the genetic sensitivity to alcohol’s effect on PCa risk [24]. It is reported that the carrier rate of hereditary disorder aldehyde dehydrogenase 2 (ALDH2) deficiency is much higher in East Asians compared to Europeans [25]. A high probability of carrying ALDH2 deficiency combined with frequent alcohol consumption may result in a greater excess risk for PCa in individuals of East Asian ancestry. 

Endorsed by the American Society of Clinical Oncology (ASCO), a vegetable-enriched diet is recommended to diminish PCa progression [26]. The potential molecular pathways include (i) antioxidants and phytochemicals from vegetables, (ii) improving the expression of cytoprotective enzymes, and (iii) promoting genomic stability [27,28]. However, a recent randomized controlled trial found a null association between vegetable-rich diets and PCa progression [28]. Given the limitations of short-term intervention and a failure to distinguish the different effects of raw and cooked vegetables, we first applied an MR approach and observed an interesting finding: consuming cooked rather than raw vegetables could decrease an 8% risk of incident PCa. Several vegetables, such as tomato, carrots, and broccoli, were reported to release more carotenoids and be better absorbed in their cooked rather than their raw status [29]. This makes the nutrition value in cooked vegetables different from that in raw counterparts, probably explaining the present finding.

Evidence regarding sexual behavior linked to PCa risk has long intrigued researchers. A meta-analysis including 22 research works showed that an increment of ten female sexual partners was associated with a 1.10-fold increase in PCa risk, while a moderate ejaculation frequency was saliently associated with a 9% lower risk of PCa [30]. A ten-year follow-up study including 31,925 male participants indicated that ejaculation frequency was inversely related to the risk of PCa [31]. The finding of our MR study supported the potential protective role of sexual behavior on PCa. Several hypotheses proposed for explanation include the prostate stagnation hypothesis, reducing prostatic intraluminal crystalloids development, hindering the metabolic switch from citrate secretion to citrate oxidation, and diminishing the stimulation of prostate epithelial cell division [4]. In fact, there may be a U-shaped relationship of sexual initiation and frequency with incident PCa risk [30]. Further research is warranted to explore the optimal initial age and frequency of sex for PCa prevention.

Inconsistency was observed between the UVMR and MVMR results in the present study. As mentioned above, a biased estimate of each single behavior could be generated in the UVMR analysis when these lifestyle behaviors were highly correlated. Based on the MVMR results, we found limited evidence suggesting the causal effect of fruit consumption and sleep behavior on PCa, which was congruent with several recent studies [8,27]. A prior MR study found an inverse relationship between accelerometer-measured PA and PCa [8]. In our UVMR analysis of the European replication sample, we also found similar evidence regarding the duration of moderate PA and PCa. Further research is needed to determine the specific conditions under which PA could protect against PCa.

Inconsistent findings for European and East Asian ancestries in our study manifested a discrepant spectrum of PCa risk factors in different regions, similar to the wide global variation in PCa incidence and genetic heterogeneity. We unfortunately did not replicate similar findings from the discovery sample in the replication sample. Thirty-three percent of PCa-SNPs were nonoverlapping between the FinnGen and PRACTICAL consortiums, which indicated a potential genetic difference between Finnish and non-Finnish Caucasians, which likely caused the inconsistent findings. In the East Asian replication sample, the low statistical power may be due to the small proportion of PCa cases within the cohort.

Some limitations of this study should be acknowledged. First, we had a limited number of SNPs for several phenotypes and, hence, the failure to detect a salient effect in these phenotypes could be due to weak instrument bias and low statistical power. Further GWAS are warranted to identify more independent genetic variants that influence lifestyle behaviors, as this would enable future analyses with sufficient power to reveal the causality between modifiable behaviors and PCa risk. Second, different lifestyle behaviors may interact in a multiplicative or additive manner. The present study was unable to perform the MR interaction analysis because individual-level data are not available. However, we have applied the multivariable MR method to account for the correlations between different behaviors and the independent effects of each. Finally, it would be interesting to explore the effects of modifiable behaviors on different stages of PCa progression. We currently could not implement such an analysis due to insufficient GWAS data.

## 5. Conclusions

In conclusion, the present MR study demonstrates that a lifetime tobacco smoking, alcohol drinking, and delayed sexual initiation may be causally and positively related to prostate cancer, whereas cooked vegetable consumption may be causally and inversely related to such cancer risk. Findings from our study broaden the evidence base for the spectrum of PCa risk factors in different ethnicities and provide public health guidance to promote healthy lifestyles for PCa prevention in different population settings.

## Figures and Tables

**Figure 1 biology-12-00673-f001:**
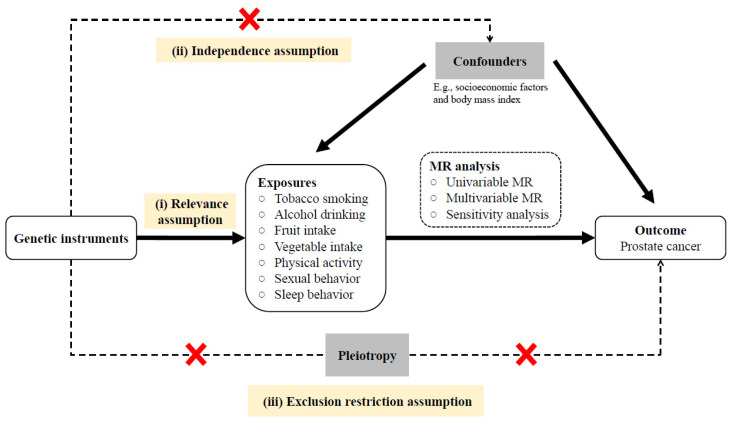
Schematic diagram of the MR design, rationale, and procedures (MR, Mendelian randomization).

**Figure 2 biology-12-00673-f002:**
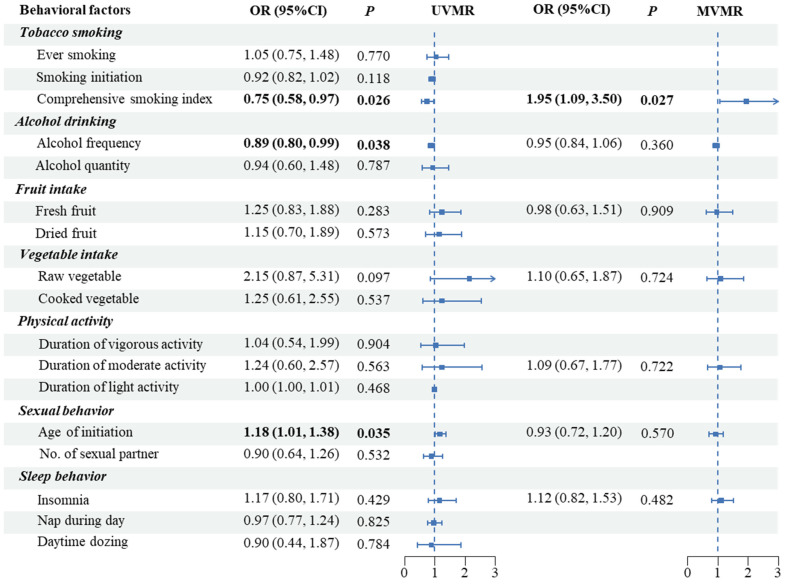
Causal effects of modifiable behaviors on prostate cancer by UVMR and MVMR in the European discovery sample (UVMR, univariable Mendelian randomization; IVW, inverse-variance weighted).

**Figure 3 biology-12-00673-f003:**
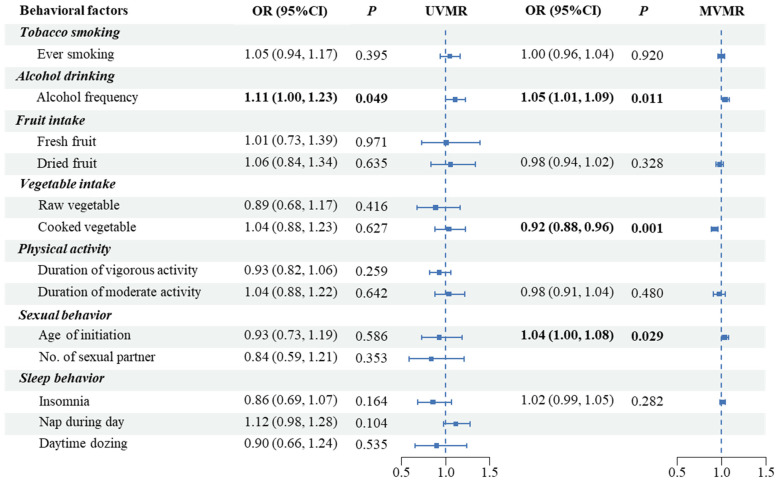
Causal effects of modifiable behaviors on prostate cancer by UVMR and MVMR in the East Asian discovery sample (MVMR, multivariable Mendelian randomization; IVW, inverse-variance weighted).

**Table 1 biology-12-00673-t001:** Details of the instruments used for proxy behaviors on prostate cancer risk in European ancestry.

Phenotype	GWAS ID	Total Population	European Discovery				European Replication			
			SNPs (n)	F-Stat	R^2^ (%)	Power (%)	SNPs (n)	F-Stat	R^2^ (%)	Power (%)
**Tobacco smoking**										
Ever smoking	ukb-b-20261	461,066	79	9.29	0.16	39	80	9.27	0.16	10
Smoking initiation	ieu-b-4877	632,802	92	172.17	2.44	>99	87	173.00	2.32	68
Lifetime smoking index	N/A	462,690	108	20.27	0.47	82	120	19.65	0.51	21
**Alcohol drinking**										
Alcohol frequency	ukb-b-5779	462,346	87	113.08	2.08	>99	97	130.57	2.40	70
Alcohol quantity	ieu-b-73	335,394	33	32.94	0.31	65	31	31.34	0.30	14
**Fruit intake**										
Fresh fruit	ukb-b-3881	446,462	49	15.60	0.17	41	53	16.65	0.18	10
Dried fruit	ukb-b-16576	421,764	36	24.08	0.21	49	39	26.64	0.23	12
**Vegetable intake**										
Raw vegetables	ukb-b-1996	435,435	20	16.52	0.08	21	22	18.85	0.09	8
Cooked vegetables	ukb-b-8089	448,651	15	16.93	0.07	20	16	24.03	0.07	7
**Physical activity**										
Duration of vigorous activity	ukb-b-13932	251,501	3	3.47	<0.01	5	3	3.47	<0.01	5
Duration of moderate activity	ukb-b-2346	343,827	3	7.07	<0.01	6	3	10.91	<0.01	5
Duration of light activity	ukb-b-8865	64,949	3	2.28	0.01	5	3	2.28	0.01	5
**Sexual behavior**										
Age of initiation	ebi-a-GCST90000047	397,338	167	16.70	0.70	94	169	16.85	0.70	27
No. of sexual partners	ukb-b-4256	378,882	59	25.70	0.39	75	59	25.41	0.39	17
**Sleep behavior**										
Insomnia	ukb-b-3957	462,341	36	8.64	0.13	33	40	12.98	0.20	11
Nap during day	ebi-a-GCST011494	452,633	96	15.20	0.32	67	95	15.88	0.34	15
Daytime dozing	ukb-b-5776	460,913	26	10.58	0.06	18	28	10.39	0.06	7

**Table 2 biology-12-00673-t002:** Details of the instruments used for proxy behaviors on prostate cancer risk in East Asian ancestry.

Phenotype	GWAS ID	Total Population	East Asian Discovery				East Asian Replication			
			SNPs (n)	F-Stat	R^2^ (%)	Power (%)	SNPs (n)	F-Stat	R^2^ (%)	Power (%)
**Tobacco smoking**										
Ever smoking	ukb-e-20160_EAS	2649	10	6.17	2.29	28	7	6.91	1.80	49
**Alcohol drinking**										
Alcohol frequency	ukb-e-1558_EAS	2658	6	18.96	4.11	46	4	20.12	2.94	70
**Fruit intake**										
Fresh fruit	ukb-e-1309_EAS	2513	6	3.41	0.81	13	5	3.85	0.76	24
Dried fruit	ukb-e-1319_EAS	2219	11	7.23	3.48	40	9	6.69	2.65	66
**Vegetables intake**										
Raw vegetables	ukb-e-1299_EAS	2365	3	2.83	0.36	9	3	2.83	0.36	14
Cooked vegetables	ukb-e-1289_EAS	2499	19	10.67	7.56	71	19	10.67	7.56	98
**Physical activity**										
Duration of vigorous activity	ukb-e-914_EAS	1373	4	5.13	1.48	20	4	5.13	1.48	42
Duration of moderate activity	ukb-e-894_EAS	1853	3	3.19	0.52	10	3	3.19	0.52	18
**Sexual behavior**										
Age of initiation	ukb-e-2139_EAS	1711	4	5.65	1.31	18	4	5.65	1.31	38
No. of sexual partners	ukb-e-2149_EAS	1559	5	9.27	2.90	34	5	9.27	2.90	69
**Sleep behavior**										
Insomnia	ukb-e-1200_EAS	2654	18	6.93	4.03	45	16	5.88	3.45	77
Nap during day	ukb-e-1190_EAS	2606	14	6.67	3.24	38	13	6.36	3.09	72
Daytime dozing	ukb-e-1220_EAS	2582	16	7.11	4.25	47	16	7.11	4.25	85

## Data Availability

All summary data were presented in the Appendix A. The study protocol, statistical analysis plan, and analytical code will be available from the time of publication in response to any reasonable request to the corresponding author.

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
