# Peer review of "Causal Effects of Modifiable Behaviors on Prostate Cancer in Europeans and East Asians: A Comprehensive Mendelian Randomization Study"

_biology, 2023, doi:10.3390/biology12050673_

Round 1

Reviewer 1 Report

I have read with interest in the manuscript, “Casual effects of modifiable behaviors on prostate cancer in Europeans and East Asians: a comprehensive Mendelian randomization study” which has been submitted to “Biology”. First of all, I congratulate this work, it is well-written manuscript with proper analysis as well discussion. The research methodology also does not raise any objections. Importantly, a sensitivity analysis was also performed, which allows for a thorough analysis of the obtained data. Moreover, extensive supplementary material has been prepared.

Author Response

Dear Reviewer,

We would like to thank you for your careful and thorough consideration of our manuscript, and your enthusiasm and recognition of this work. 

[1] Reviewer’s comment: First of all, I congratulate this work, it is well-written manuscript with proper analysis as well discussion. The research methodology also does not raise any objections. Importantly, a sensitivity analysis was also performed, which allows for a thorough analysis of the obtained data. Moreover, extensive supplementary material has been prepared.

【Our Response】Thank you very much for your support and recognition of this manuscript.

Best regards.

Reviewer 2 Report

Dear Authors, I have read with interest your article entitled “Causal effects of modifiable behaviors on prostate cancer in Europeans and East Asians: a comprehensive Mendelian randomization study”. 

I think it adds some useful information in the panorama of the epidemiology of prostate cancer.

Here reported are some limitations to correct:

·       English is not always fluent and sometimes terms are obsolete; please check with a native speaker

·       Statistical analyses are complex to read for average Urologists

·       Please consider citing the following pertinent paper

o   Minerva Urology and Nephrology 2021 August;73(4):442-51. The emerging landscape of tumor marker panels for the identification of aggressive prostate cancer: the perspective through bibliometric analysis of an Italian translational working group in uro-oncologyMatteo FERRO, Giuseppe LUCARELLI, Ottavio de COBELLI, Francesco DEL GIUDICE, Gennaro MUSI, Francesco A. MISTRETTA, Stefano LUZZAGO, Gian M. BUSETTO, Carlo BUONERBA, Alessandro SCIARRA, Simon CONTI, Angelo PORRECA, Michele BATTAGLIA, Pasquale DITONNO, Matteo MANFREDI, Cristian FIORI, Francesco PORPIGLIA, Daniela TERRACCIANO

Author Response

Dear Reviewer,

We would like to thank you for your careful and thorough consideration of our manuscript, and your recognition of this work. We have modified our manuscript accordingly. With the help of reviewers, we believe that our manuscript has been largely improved. Please also refer to the following point-to-point response to the reviewer’s comments:

[1] Reviewer’s comment: English is not always fluent and sometimes terms are obsolete; please check with a native speaker.

【Our Response】We have sent our manuscript to the grammatical and language review service by a native-English editor. An Editing Certificate was uploaded as supporting material.

[2] Reviewer’s comment: Statistical analyses are complex to read for average Urologists.

【Our Response】Thank you for your suggestion. We have revised the statistical analyses section using plain language that can be readable for general urologists. The sophisticated statistical method was detailed in Appendix 1 for those interested researchers reference.

[3] Reviewer’s comment: Please consider citing the following pertinent paper.

【Our Response】We have cited this pertinent paper properly.

Yours sincerely.

Reviewer 3 Report

Dear authors of “Causal effects of modifiable behaviors on prostate cancer in  Europeans and East Asians: a comprehensive Mendelian randomization study”,

Thanks for your contribution on this field. This is an interesting article, well written and organized.

Indeed, my major concern is about the items compare between the 2 population: European ancestry vs East Asian ancestry. They should be at least the same.

Example:

> Tobacco smoking: Smoking initiation, Lifetime smoking index vs Tobacco smoking: Ever smoking.

> Alcohol quantity and Duration of light activity absent in East Asian ancestry.

Also, Nap during day is not really associated with sleeping behavior. Maybe the authors must create a major item Food intake and put together Fruit, Vegetable and Nap.

Manuscript text must be corrected according to the made modifications.

Hoping the authors can address those raised points,

Best regards.

Author Response

Dear Reviewer,

We would like to thank you for your careful and thorough consideration of our manuscript, and your recognition of this work. We have modified our manuscript accordingly. With the help of reviewers, we believe that our manuscript has been largely improved. Please also refer to the following point-to-point response to the reviewer’s comments:

[1] Reviewer’s comment: My major concern is about the items compare between the 2 population: European ancestry vs East Asian ancestry. They should be at least the same. Example: Tobacco smoking: Smoking initiation, Lifetime smoking index vs Tobacco smoking: Ever smoking. Alcohol quantity and Duration of light activity absent in East Asian ancestry.

【Our Response】Thank you for your insightful comments. We have tried our utmost to collect the same 13 items between two populations. We have added the “Ever smoking” phenotype in the European population consistent with that in the East Asian population. We have added the corresponding analysis and modified the results in the Tables, Figures, and the text. Given the limitation of the open GWAS dataset, we fail to collect other 4 items absent in East Asian ancestry. The absent items can be further examined in the future when data are applicable. As all the analyses in this research were conducted separately in two populations, and the interpretation of the results were based on the two different populations, we believed that such limitation would not change our conclusion.     

[2] Reviewer’s comment: Also, Nap during day is not really associated with sleeping behavior.

【Our Response】We agreed that “nap during day” may be disputable in terms of the sleeping behavior. As a matter of fact, nap is included as an item in several sleep-related scales such as the Quebec Sleep Questionnaire (QSQ) (1), and Sleep Scale from the Medical Outcomes Study (MOS-Sleep) (2). According to the definition, a nap is referred to a ‘short sleep’ that is considerably shorter than an individual’s normal sleep episode, which belongs to sleep behavior from a general and public health perspective (3). Thus, we retain “nap” in the sleep behavior group, to distinguish the effect between nap and general sleep. In addition, we add another phenotype “daytime dozing” that is more related to sleeping behavior and is most commonly used in sleep research and clinical settings (4). The corresponding analysis and modification can be found in the Tables, Figures, and the text.

References:

(1) Lacasse Y, et al. A new standardised and self-administered quality of life questionnaire specific to obstructive sleep apnoea. Thorax. 2004,59(6):494-9. DOI: 10.1136/thx.2003.011205;

(2)Shahid, A., et al. Medical Outcomes Study Sleep Scale (MOS-SS). Springer, New York, NY, 2011. https://doi.org/10.1007/978-1-4419-9893-4_50.

(3) Lastella M, et al. To Nap or Not to Nap? A Systematic Review Evaluating Napping Behavior in Athletes and the Impact on Various Measures of Athletic Performance. Nat Sci Sleep. 2021,13:841-862. Doi: 10.2147/NSS.S315556.

(4) Kendzerska TB, et al. Evaluation of the measurement properties of the Epworth sleepiness scale: a systematic review. Sleep Med Rev. 2014,18(4):321-31. DOI: 10.1016/j.smrv.2013.08.002.

[3] Reviewer’s comment: Maybe the authors must create a major item Food intake and put together Fruit, Vegetable and Nap.

【Our Response】Thanks for your advice. We agreed that it would be an interesting topic to discuss in the future. However, as a MR study, our analyses were based on the summary data from the published GWASs. The variables and magnitude of association were already established. Therefore, we are unable to create a new item (including fruit, vegetable and nap together) which has not been reported by the existing studies. In our results, we have also performed a multivariable MR analysis adjusting for different behaviors. We believe that the independent effects of each behavior are more credible and robust in the multivariable models.

[4] Reviewer’s comment: Manuscript text must be corrected according to the made modifications.

【Our Response】We have added “Ever smoking” and “Daytime dozing” phenotypes according to your prior suggestions. The corresponding analysis and modifications were made in the text, Figures, Tables, and supplementary materials.

Yours sincerely.

Round 2

Reviewer 3 Report

Dear authors,

Thanks for taking in consideration my comments.

All my apologies for nap during the day.

I was overloaded with work and misunderstood the point.

Congrats for this nice contribution in PCa field.

All the best.